# The impact of management compliance attention and board faultlines strength on ESG performance: Evidence from Chinese listed companies

Yong Jiang, Fei Han*

School of Accounting, Shandong Management University, Jinan, Shandong, China

* hanfei246@163.com

## Abstract

Against the backdrop of global sustainable development goals and the rapid evolution of ESG practices in China, this study empirically examines how management compliance attention and board faultlines strength affect ESG performance using data from Chinese A-share listed companies spanning 2010–2022. Using textual analysis of annual reports to measure compliance attention, we employ OLS (Ordinary Least Squares) regression, moderation, and mediation analyses. The findings reveal three key results: (1) Management compliance attention is positively associated with ESG performance ($\beta = 0.041$, $p < 0.01$); (2) Board faultlines strength significantly amplifies this positive impact ($\beta = 0.114$, $p < 0.01$); (3) Organizational resilience partially mediates the relationship between management compliance attention and ESG performance, as confirmed by a Sobel test ($z = 3.089$, $p = 0.002$). Heterogeneity analysis shows that state-owned enterprises exhibit more pronounced compliance-driven ESG improvements, whereas private firms rely on board diversity to enhance such effects. This study contributes to the literature by integrating institutional theory, cognitive theory, upper echelons theory, and dynamic capabilities theory, demonstrating that management compliance attention boosts ESG performance through organizational resilience, with board faultlines acting as a critical moderator. Practical implications include recommendations for firms to strengthen compliance cultures, foster diverse boards, and build organizational resilience, while policymakers should enhance compliance frameworks and ESG disclosure standards to promote sustainable corporate practices.

## 1. Introduction

Against accelerated global pursuit of sustainable development goals, ESG (Environmental, Social, and Governance) has become a core dimension for evaluating corporate strategic transformation. By 2023, global ESG assets reached $82 trillion,

**Data availability statement:** All relevant data are within the manuscript and its Supporting Information files.

**Funding:** The author(s) received no specific funding for this work.

**Competing interests:** The authors have declared that no competing interests exist.

accounting for 25% of globally managed assets [1]. ESG's role in corporate sustainability has emerged as a central topic in both academic and practical discourse. It reflects how companies fulfill social responsibilities to investors and meet public expectations [2], serving as a key indicator for addressing environmental and social risks and a core basis for capital markets to assess long-term value [3]. As of 2023, 1,738 Chinese A-share listed companies had released ESG reports, a 22.14% increase from 2022, signaling growing market responsiveness to ESG [4]. However, many Chinese listed companies lack clear understanding of ESG's strategic value, leading to insufficient intrinsic motivation to improve performance [5]. ESG performance varies significantly across Chinese firms, shaped by both institutional pressures and corporate governance structures. Some companies enhance ESG performance through green innovation and governance reforms [6], while others see ESG more effectively curb management misconduct when information transparency is low [7].

Corporate compliance strengthens organizational risk management and competitive positioning. Senior management ensures regulatory adherence by prioritizing external compliance monitoring and refining internal governance frameworks [8]. Management compliance attention—defined as proactive focus on regulatory requirements—directly results from corporate compliance policies. Recent challenges faced by Chinese multinational firms (e.g., Huawei and TikTok) in navigating cross-border regulations have spurred systematic improvements to their compliance systems. Aligning with international standards and establishing standardized frameworks has enhanced their adaptive capabilities in global markets [9].

Team faultlines provide a foundational framework for analyzing board diversity and effectiveness [10]. Coined by Lau & Murnighan (1998), faultlines refer to hypothetical divisions within groups that split members into internally homogeneous subgroups based on multi-dimensional attributes (e.g., demographics, expertise, values) [11]. Their impact on team performance depends on context, with environmental conditions determining positive or negative effects [12]. Strong TMT faultlines often increase performance variability due to intensified subgroup conflicts and coordination challenges [13]. As the apex of corporate governance, boards play a pivotal role in strategic resource allocation, fostering innovation, and cultivating external networks [8,14]. Board characteristics significantly influence ESG performance, highlighting the need to examine faultline dynamics in this context [15]. This study investigates how board faultlines strength moderates the relationship between management compliance attention and ESG performance among Chinese A-share listed companies, integrating institutional and upper echelons theories to unpack contextual mechanisms.

China's unique context combines policy pressures (e.g., government environmental assessments) and market mechanisms. Existing literature explores various influences on ESG performance but overlooks the synergy between management compliance attention and board structure. This paper addresses two key questions: 1. How does management compliance attention affect ESG performance? 2. What role does board faultlines strength play in this relationship? This study aims to uncover the mechanism of "management compliance attention—board faultlines

strength—ESG performance", provide a differentiated theoretical framework for ESG governance in emerging markets, and respond to both global ESG trends and China's institutional context.

This study makes three main contributions. First, it operationalizes management compliance attention beyond abstract institutional responses, constructing a quantitative indicator through annual report textual analysis. It empirically tests the positive impact of compliance attention on ESG performance, breaking from traditional studies focused solely on financial compliance or institutional pressures. Second, it introduces "board faultlines strength", revealing how cognitive conflicts and integration from internal subgrouping amplify compliance attention's effect on ESG performance through enhanced supervision and strategic innovation. Third, it identifies organizational resilience as a mediating variable, showing that compliance attention enhances ESG performance through resilience. Further, it distinguishes heterogeneous impacts by ownership and market competition: state-owned companies rely on institutional pressures to drive the compliance-ESG chain, while private firms depend more on board diversity to activate compliance effectiveness. This provides a differentiated perspective for corporate governance and sustainable development research in emerging markets.

## 2. Literature review and hypothesis development

### 2.1. Management compliance attention and ESG performance

Drawing on Institutional Theory and Cognitive Theory, this subsection explores how management compliance attention affects ESG performance, offering theoretical support to understand micro-dynamics of sustainable development in Chinese companies.

Management promotes ESG performance by enhancing corporate governance and sustainable development [16]. External drivers, such as public environmental concern [17,18] and investor attention [19,20], also improve ESG performance. Internally, compliance and ethical management strengthen the positive effect of CSR (Corporate Social Responsibility) on financial performance [21]. ESG strategy implementation requires prioritizing social compliance in governance [22]. However, ESG scores correlate with voluntary ESG disclosure quantity but not actual compliance records [23]. Asia's current corporate compliance levels remain insufficient to ensure adherence to ESG regulations [24]. Additionally, socially responsible strategies expose firms to higher compliance costs and risks [25]. Despite these insights, literature remains fragmented: theories lack integration, and measurements often rely on proxy variables. Research also neglects China's contextual nuances, mediating mechanisms, and board faultlines' potential moderation of the compliance-ESG relationship. This creates opportunities for more integrated, context-specific studies with robust measurement frameworks.

**2.1.1. Institutional theory perspective—Legitimacy-driven by external pressures.** Institutional theory posits that the survival and development of organizations depend on their adaptation to the external institutional environment, which provides regulative, normative, and cognitive structures and activities that provide stability and meaning to social behavior [26]. From an institutional theory perspective, regulations, as mandatory institutions, directly drive the enhancement of corporate social responsibility [27].

Managers' perceptions and beliefs significantly shape behavioral patterns and decision-making processes [28]. Organizational adaptation to institutions forms constraints and guidance through three dimensions—regulative, normative, and cognitive—directly prompting management to translate compliance attention into ESG practices. Regulative pressure imposes "hard constraints" through mandatory regulations, compelling management to incorporate compliance requirements into strategic agendas. This pressure directly enhances management's attention to ESG-related compliance. From the normative pressure perspective, "soft constraints" such as industry conventions and social expectations influence management decisions. To gain recognition from investors or consumers, management proactively shifts compliance focus toward ESG-related domains. Cognitive pressure leads to internalization of compliance cognition into management's belief system, increasing ESG investment and driving performance improvements.

**2.1.2. Cognitive theory perspective—Transformation and amplification of institutional pressures.** Institutional pressures merely provide a "starting point" for compliance attention, while management's cognitive processes determine whether such pressures can be transformed into substantive ESG performance. Cognitive theory suggests that management's perceptions and beliefs significantly influence actions and decision-making [29]. Management faces limited cognitive resources and prioritizes institutional signals aligning with their cognitive frameworks. Institutional pressures often exist in abstract forms. Management converts them into operational ESG indicators through cognitive processes of "interpretation, decomposition, and meaning attribution". Strong cognitive commitment to ESG principles prompts management to promote internal controls and risk management consistent with sustainable development goals. Good internal controls and risk management practices improve ESG information disclosure quality and transparency [30]. Prioritizing compliance helps managers ensure companies meet basic legal and ethical standards, avoiding negative impacts on ESG performance. Management's cognitive commitment to sustainability drives proactive strategies beyond mere compliance, fostering innovation and improved stakeholder engagement [29].

Under external institutional pressures and internal cognitive transformation, management's focus on environmental compliance manifests in strategic interpretation of institutional signals—such as carbon pricing policies, emission standards, and ecological protection mandates. Compliance-driven strategic adjustments materialize in three key domains: (1) firms increase R&D investments in clean energy and energy-efficient technologies to meet regulatory thresholds, building environmental technological advantages; (2) they adopt systems like ISO 14001 and implement circular economy models to reduce production-related emissions and resource waste; (3) companies extend compliance requirements to supply chains through meta-regulation, forming a "compliance ecosystem network" [31]. Management compliance attention in the social dimension responds to institutional mandates—such as labor regulations, consumer rights protection, and community obligations—with core objective of establishing social legitimacy by addressing diverse stakeholder needs [32]. Moreover, managerial focus on statutes like the Labor Contract Law drives companies to enhance employee compensation structures, strengthen occupational health and safety protocols, and cultivate inclusive organizational cultures. These actions mitigate labor dispute risks and fortify organizational resilience by fostering employee psychological ownership [33]. Management compliance attention at the corporate governance level centers on adherence to company law, governance codes, and regulatory mandates. It aims to enhance governance structures through institutional adaptation and reduce agency costs [34]. Managerial efforts toward governance compliance begin with structural optimizations such as establishing dedicated ESG committees, increasing the proportion of independent directors, and forming a "institutions-processes-supervision" three-dimensional risk control framework. Additionally, the communication effect of social responsibility disclosures reduces information asymmetry and enhances corporate transparency. Complying with corporate governance codes reduces the likelihood of financial distress [35].

In summary, institutional theory provides the legitimacy rationale for "why management should pay attention" to compliance, while cognitive theory explains the practical logic of "how to pay attention". The impact of management compliance attention on ESG performance stems not from external pressures or internal cognition alone but from combined institutional constraints and cognitive agency. Institutional pressures ensure the "necessity" of compliance attention, while cognitive processing ensures its "effectiveness". Together, they drive systematic improvements in environmental, social, and governance performance.

As illustrated in Fig 1, management compliance attention influences ESG performance through three layers.

Therefore, the following hypothesis is proposed:

H1: Management compliance attention is positively associated with ESG performance.

## 2.2. Board faultlines strength and ESG performance

Drawing on upper echelons theory, this subsection analyzes how board faultlines strength influences ESG performance through information diversity and supervisory effectiveness. It addresses existing research gaps in examining dynamic effects of board structure, offering new perspectives for optimizing corporate governance structures.

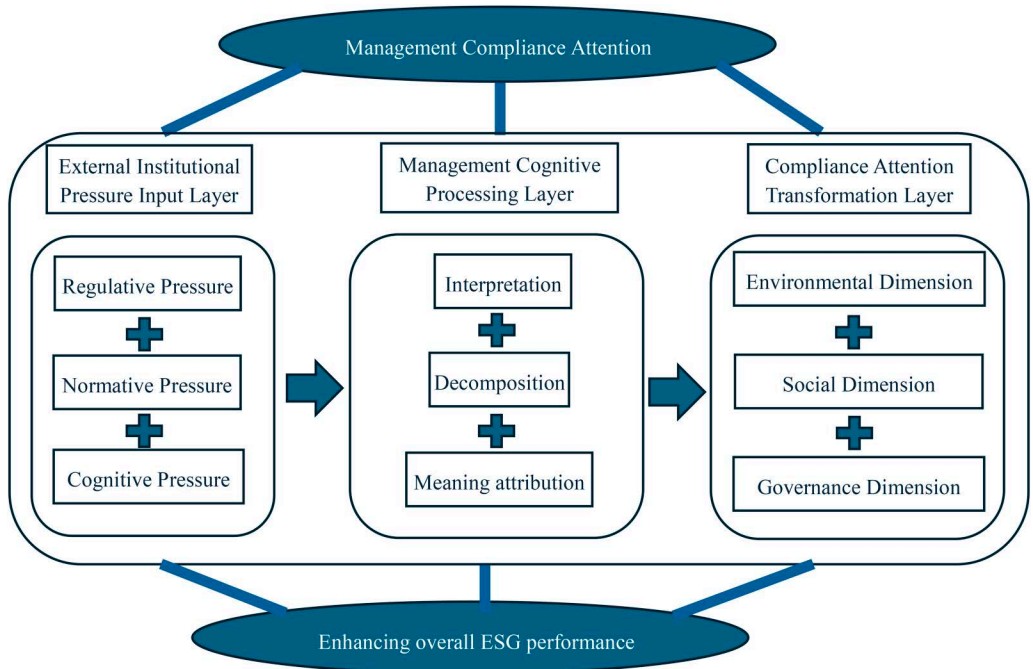

**Fig 1. Theoretical framework.**

Upper echelons theory posits that board members' demographic characteristics influence cognitive patterns and strategic preferences [36]. Board faultlines strength reflects the alignment of members' diverse attributes [11]. Members with differing characteristics form distinct subgroups. For example, directors with environmental expertise often prioritize sustainability [37], whereas financial or legal experts emphasize risk control and compliance. These cognitive differences are amplified through faultline mechanisms, prompting boards to integrate multi-dimensional information into ESG strategy formulation. Strong faultlines foster information diversity, boosting ESG strategy innovation and risk identification [38]. This cognitive foundation helps firms navigate institutional pressures like China's "dual carbon" goals,

Board faultlines strength affects ESG performance through two mechanisms. One mechanism involves professional complementarity and supervisory effectiveness. Subgroup differences from faultlines enable cross-validation in board decisions on environmental investments and social responsibility projects—for instance, female directors focus more on employee rights and community relations [39,40], financial experts optimize resource allocation, and legal experts ensure compliance [41]. This division of labor enhances ESG strategy implementation. Another mechanism involves legitimacy acquisition and stakeholder responsiveness. High-faultline boards integrate stakeholder perspectives (such as consumers and regulators), aligning ESG practices with social norms. Directors with varied tenures balance short-term objectives and long-term sustainability [42]. Moderate heterogeneity enhances strategic resilience in environmental innovation, social governance, and compliance. This cognitive diversity ultimately improves ESG performance.

Therefore, the following hypothesis is proposed:

H2: Board faultlines strength is positively associated with ESG performance.

## 2.3. Management compliance attention, board faultlines strength and ESG performance

Board faultlines, as a fundamental governance mechanism, simultaneously embody structural heterogeneity and cognitive diversity within management teams. By integrating institutional theory and upper echelons theory, this study

reveals how board faultlines structurally amplify the translation of compliance attention into ESG performance through governance-cognition interactions. Board faultlines foster innovation and organizational change [37]. Active oversight from board faultlines incentivizes management to prioritize compliance. Faultlines from cognitive or functional diversity introduce multifaceted viewpoints to board deliberations, enabling more robust and adaptive compliance strategies. Interplay of divergent thinking in boardrooms spurs innovative solutions to complex challenges, fostering intra-organizational communication and collaboration [43]. Cultivating inclusive decision-making environments, boards with pronounced faultlines enhance depth and balance of compliance discussions, ensuring strategies reflect comprehensive risk assessments and stakeholder needs. Moreover, strong board faultlines serve as a mechanism for representing diverse stakeholder interests [44]. Expanded stakeholder representation drives greater ESG engagement, compelling management to integrate compliance into strategic initiatives. The synergy of three mechanisms—enhanced monitoring, cognitive diversity, and groupthink mitigation—elevates compliance efficacy, which subsequently bolsters ESG performance through optimized risk control, stakeholder engagement, and sustainability goal internalization. Thus, this study proposes the following hypothesis:

H3: Board faultlines strength strengthens the positive impact of management compliance attention on ESG performance.

## 3. Research design

### 3.1. Data collection

The initial data were obtained from the China Stock Market and Accounting Research (CSMAR) database. Implemented since July 1, 2009, China's Basic Standards for Enterprise Internal Control mandates corporate internal controls to ensure legal and compliant operations, and because ESG disclosure rates among listed companies were low before 2010 and data from earlier years failed to sufficiently support empirical analysis, this study selects A-share listed companies from 2010 to 2022 as the research sample, excluding financial institutions, firms labeled as ST or *ST, delisted or suspended firms, and firms with missing values. Financial firms operate under sector-specific regulations (e.g., Commercial Bank Law), making their compliance systems and ESG metrics incomparable to non-financial firms; ST and *ST firms are typically labeled due to consecutive losses, financial fraud, and other reasons, and including them in the sample would introduce extreme values that affect the robustness of regression results. The final sample reflects a manufacturing-dominated, multi-industry composition, and based on China's "dual carbon" goals and the fact that ESG disclosure systems primarily target non-financial entities, the research sample selected in this paper aligns with policy focus areas. Additionally, we excluded extreme values at the 1st and 99th percentiles. After these procedures, a final sample of 23,973 firm-year observations was obtained. To reduce the probability of heteroscedasticity in the linear regression model, we applied natural logarithm transformations to the values of the dependent variable, independent variable, and some control variables. This study did not involve human participants, animal subjects, or primary data collection, thus ethics review was not applicable.

### 3.2. Variables

**3.2.1. Dependent variable.** ESG performance serves as the dependent variable. We utilize ESG scores from the CNRDS (China Research Data Services Platform), which constructs composite ratings by integrating 9 social indicators, 10 governance indicators, 6 diversity indicators, 10 employee relationship indicators, 10 environmental indicators, and 13 product-related key performance indicators. Scores range from 0 to 100 percent, with natural logarithm transformation applied to measure overall ESG performance.

**3.2.2. Independent variable.** Management compliance attention serves as the independent variable. Following Davenport & Beck (2001), attention is defined as "mental concentration on specific information accompanied by action decisions" [45]. Management prioritizes issues within their supervisory scope while excluding irrelevant matters [46]. In the compliance domain, management's proactive attention essentially represents selective attention to institutional

pressures, requiring systematic keyword systems to capture the focus direction of this attention. Regulative, normative, and cultural-cognitive elements constitute institutional foundations, shaping behaviors through distinct mechanisms [26]. Following prior research on corporate compliance [47,48], this study extracts keywords from the "Management Discussion and Analysis" section of annual reports to compile a compliance-focused keyword list. We then measured management compliance attention by counting the frequency of these keywords and taking the natural logarithm of the frequency after adding one. Specifically, five keywords, namely "compliance, legitimacy, regulations, laws, and rules" were categorized as the "compliance foundation". These terms reflect the management's response to regulatory pressures and their pursuit to mitigate legal risks. Four keywords, including "standards, auditing, supervision, and risk management" were grouped under "implementation mechanisms". They signify the management's reaction to normative pressures and their efforts in board governance and agency cost control. The two keywords, "information security" and "data protection" were identified as compliance focal points in the "emerging fields". They mirror the management's understanding of the digital economy, and their selection demonstrates a strategic choice to proactively address technological regulations [49].

**3.2.3. Moderating variable.** Drawing on the theoretical framework of Lau & Murnighan (1998) [11], faultlines arise from the alignment of multiple attributes among group members, leading to the formation of homogeneous subgroups and potential intergroup divisions [50]. The process involves five key steps: First, selecting relevant attributes that may influence subgroup formation, such as demographic and task-related characteristics. Second, categorizing these attributes into meaningful discrete categories to reflect cognitive perceptions (e.g., age groups or job types). Third, calculating internal alignment within subgroups where higher values indicate greater homogeneity. Fourth, measuring cross-subgroup alignment to assess attribute overlap between subgroups, with higher values indicating more shared attributes. Finally, combining these into a Faultline Strength (FLS) index through the formula:

$$\text{FLS} = \text{IA} \times (1 - \text{CGAI}) \tag{1}$$

IA is the internal alignment and CGAI is the cross-subgroup alignment index. This framework quantifies how attribute patterns create potential divides, offering a systematic measure distinct from traditional diversity indices.

In this paper, we operationalize board demographic characteristics—gender, age, education, and tenure—to construct board faultline indicators as follows:

$$\text{FLS}_1 = \frac{\text{IA}_{12} \times (1 - \text{CGAI}_{12}) + \text{IA}_{13} \times (1 - \text{CGAI}_{13}) + \text{IA}_{14} \times (1 - \text{CGAI}_{14})}{3} \tag{2}$$

$$\text{FLS}_2 = \frac{\text{IA}_{21} \times (1 - \text{CGAI}_{21}) + \text{IA}_{23} \times (1 - \text{CGAI}_{23}) + \text{IA}_{24} \times (1 - \text{CGAI}_{24})}{3} \tag{3}$$

$$\text{FLS}_3 = \frac{\text{IA}_{31} \times (1 - \text{CGAI}_{31}) + \text{IA}_{32} \times (1 - \text{CGAI}_{32}) + \text{IA}_{34} \times (1 - \text{CGAI}_{34})}{3} \tag{4}$$

$$\text{FLS}_4 = \frac{\text{IA}_{41} \times (1 - \text{CGAI}_{41}) + \text{IA}_{42} \times (1 - \text{CGAI}_{42}) + \text{IA}_{43} \times (1 - \text{CGAI}_{43})}{3} \tag{5}$$

Equation (2) represents the gender faultline strength of the board (FLS$_1$), where IA$_{12}$ is the internal synergy indicator based on gender and age, IA$_{13}$ is the internal synergy indicator based on gender and education, and IA$_{14}$ is the internal synergy indicator based on gender and tenure. CGAI$_{12}$ is the cross-group synergy indicator based on gender and age, CGAI$_{13}$ is the cross-group synergy indicator based on gender and education, and CGAI$_{14}$ is the cross-group synergy indicator based on gender and tenure. Equations (3) to (5) represent the age faultline strength (FLS$_2$), education faultline

strength (FLS$_3$), and tenure faultline strength (FLS$_4$) of the board respectively, with the same composition as FLS$_1$. The average of FLS$_1$, FLS$_2$, FLS$_3$, and FLS$_4$ is taken to obtain the board faultlines strength indicator Board Faultlines.

In addition to the above method, this paper constructs another board faultlines strength indicator in the following way, which is used as a substitute variable for robustness testing in the subsequent text. The average of NFLS$_1$, NFLS$_2$, NFLS$_3$, and NFLS$_4$ is taken to obtain a new board faultline indicator NBoard Faultlines.

$$NFLS_1 = \frac{IA_{12}+IA_{13}+IA_{14}}{3} \times (1 - \frac{CGAI_{12}+CGAI_{13}+CGAI_{14}}{3}) \tag{6}$$

$$NFLS_2 = \frac{IA_{21}+IA_{23}+IA_{24}}{3} \times (1 - \frac{CGAI_{21}+CGAI_{23}+CGAI_{24}}{3}) \tag{7}$$

$$NFLS_3 = \frac{IA_{31}+IA_{32}+IA_{34}}{3} \times (1 - \frac{CGAI_{31}+CGAI_{32}+CGAI_{34}}{3}) \tag{8}$$

$$NFLS_4 = \frac{IA_{41}+IA_{42}+IA_{43}}{3} \times (1 - \frac{CGAI_{41}+CGAI_{42}+CGAI_{43}}{3}) \tag{9}$$

**3.2.4. Control variables.** Companies with a large number of employees may have more resources to invest in ESG practices; firms may adjust ESG investments due to financial pressures; high-growth companies may focus more on short-term performance and overlook long-term ESG goals; high-dividend firms may face a trade-off between shareholder returns and ESG investments; companies with high asset utilization efficiency may be better at integrating resources to achieve ESG objectives; the proportion of independent directors may directly influence the formulation and supervision of ESG strategies; and executive incentive mechanisms may affect their commitment to ESG. Therefore, this paper controls the following variables: Employee, Lev (leverage ratio), Revgrowth (sales revenue growth rate), Div_ratio (dividend payout ratio), Taturn (total asset turnover), Indep_ratio (proportion of independent directors), and Top3_pay (compensation of top three executives). Control variables and other specific variables are illustrated in Table 1.

### 3.3. Empirical models

Building on Fama & French (1998) [51], we adopt the empirical framework of Fatemi et al. (2018), and Chen et al. (2023) [52,53]to specify the following model for testing the proposed hypotheses:

$$\begin{aligned}
ESG_{it} =\ & \beta_0 + \beta_1 Compliance\ Attention_{it} + \beta_2 Board\ Faultlines_{it} \\
& + \beta_3 Compliance\ Attention_{it} \times Board\ Faultlines_{it} + \beta_4 Employee_{it} \\
& + \beta_5 Lev_{it} + \beta_6 Revgrowth_{it} + \beta_7 Div\_ratio_{it} + \beta_8 Taturn_{it} + \beta_9 Indep\_ratio_{it} \\
& + \beta_{10} Top3\_pay_{it} + \sum Industry + \sum Year + \varepsilon_{it}
\end{aligned} \tag{10}$$

## 4. Results

### 4.1. Descriptive statistics

Table 2 reports descriptive statistics for all variables. Before taking the natural logarithm, the average ESG performance (ESG) is 24.582, the median is 24.143, and the standard deviation is 1.495; within the ESG score range of 0–100, these figures indicate that the overall ESG performance of sample companies is relatively low and that certain differences exist among companies. The pre-log average of Compliance Attention is 7.243, the median is 7.001, and the standard deviation is 2.643, indicating that there exists certain level of compliance attention in management with significant heterogeneity across companies. The average value of Board Faultlines is 0.088, the median is 0.093, the standard deviation is 0.052, and the range is 0 to 0.214, reflecting that the overall board faultlines strength is relatively low, and that significant differences in faultlines strength based on demographic characteristics exist across boards. Descriptive statistics for control variables align with prior research, and detailed interpretations are omitted here for brevity.

**Table 1. Variable description.**

| Variable type | Variable name | Definition |
|---|---|---|
| Dependent variable | ESG | Natural logarithm of ESG score |
| Independent variable | Compliance Attention | Natural logarithm of (number of management compliance concern key words frequency+1) |
| Moderating variable | Board Faultlines | Refer to details in "3.2.3 Moderating variable" on measurement |
| Control variables | Employee | Natural logarithm of number of employees |
| | Lev | Total debt divided by total assets |
| | Revgrowth | (Current sales revenue-Prior sales revenue)/Prior sales revenue |
| | Div_ratio | Dividend payout ratio |
| | Taturn | Sales revenue divided by average total assets |
| | Indep_ratio | Number of independent directors divided by number of board members |
| | Top3_pay | Natural logarithm of top 3 executives compensation |
| | Year | 2010-2022 |
| | Industry | China Securities Regulatory Commission 2012 industry classification |

**Table 2. Descriptive statistics.**

| Variable | Obs | Mean | Std.Dev. | Median | Min | Max |
|---|---|---|---|---|---|---|
| ESG | 23,973 | 3.202 | 0.402 | 3.184 | 2.070 | 4.061 |
| Compliance Attention | 23,973 | 1.980 | 0.972 | 1.946 | 0.000 | 4.290 |
| Board Faultlines | 23,973 | 0.088 | 0.052 | 0.093 | 0.000 | 0.214 |
| Employee | 23,973 | 7.712 | 1.204 | 7.617 | 5.136 | 11.142 |
| Lev | 23,973 | 0.402 | 0.195 | 0.394 | 0.051 | 0.845 |
| Revgrowth | 23,973 | 0.207 | 0.372 | 0.136 | −0.406 | 2.330 |
| Div_ratio | 23,973 | 0.274 | 0.303 | 0.214 | 0.000 | 1.802 |
| Taturn | 23,973 | 0.630 | 0.402 | 0.540 | 0.106 | 2.426 |
| Indep_ratio | 23,973 | 0.375 | 0.053 | 0.333 | 0.333 | 0.571 |
| Top3_pay | 23,973 | 14.514 | 0.713 | 14.487 | 12.794 | 16.524 |

## 4.2. Correlated coefficient matrix analysis

Table 3 reports the Pearson correlation coefficient matrix for compliance attention, ESG performance and other variables. The correlation coefficient between Compliance Attention and ESG is −0.064 (significant at the 1% level), which contradicts H1. This counterintuitive result is likely due to the exclusion of control variables in the bivariate analysis, necessitating further multivariate testing. The correlation coefficient between Board Faultlines and ESG is 0.058 (significant at the 1% level), which preliminarily supports H2. The maximum correlation coefficient in the matrix is 0.414, indicating no severe multicollinearity—providing a robust foundation for subsequent regression analysis.

## 4.3. Baseline regression

Table 4 presents the results of four models covering the period from 2010 to 2022. In models 1–4, the R-squared values increase from 0.451 to 0.453, indicating that the model fitness gradually improves as variables are added, and the F-values of Model 1 to Model 4 which are all greater than 200 indicate that all models fit well. In Model 4, management compliance attention has a significant positive effect on ESG performance ($\beta = 0.041$, $p < 0.01$), meaning that for each unit

**Table 3. Correlation matrix.**

| | 1 | 2 | 3 | 4 | 5 | 6 | 7 | 8 | 9 | 10 |
|---|---|---|---|---|---|---|---|---|---|---|
| 1. ESG | 1.000 | | | | | | | | | |
| 2. Compliance Attention | −0.064*** | 1.000 | | | | | | | | |
| 3. Board Faultlines | 0.058*** | −0.060*** | 1.000 | | | | | | | |
| 4. Employee | 0.142*** | −0.013** | −0.024*** | 1.000 | | | | | | |
| 5. Lev | 0.000 | −0.010 | −0.081*** | 0.414*** | 1.000 | | | | | |
| 6. Revgrowth | −0.037*** | 0.046*** | 0.015** | −0.014** | 0.068*** | 1.000 | | | | |
| 7. Div_ratio | 0.008 | −0.021*** | 0.020*** | −0.005 | −0.173*** | −0.107*** | 1.000 | | | |
| 8. Taturn | 0.038*** | 0.054*** | −0.070*** | 0.266*** | 0.196*** | 0.065*** | −0.011* | 1.000 | | |
| 9. Indep_ratio | 0.015** | −0.037*** | −0.060*** | −0.006 | −0.011* | 0.001 | −0.007 | −0.021*** | 1.000 | |
| 10. Top3_pay | 0.144*** | −0.154*** | 0.014** | 0.349*** | 0.125*** | 0.021*** | −0.001 | 0.094*** | 0.021*** | 1.000 |

Note: *** $p < 0.01$, ** $p < 0.05$, * $p < 0.1$.

**Table 4. Baseline regression.**

| VARIABLES | Model 1 ESG | Model 2 ESG | Model 3 ESG | Model 4 ESG |
|---|---|---|---|---|
| Compliance Attention | 0.042*** | | 0.042*** | 0.041*** |
| | (16.330) | | (16.172) | (16.002) |
| Board Faultlines | | 0.356*** | 0.344*** | 0.344*** |
| | | (9.184) | (8.903) | (8.908) |
| Compliance Attention × Board Faultlines | | | | 0.114*** |
| | | | | (2.882) |
| Employee | 0.041*** | 0.041*** | 0.040*** | 0.040*** |
| | (19.110) | (19.253) | (18.971) | (19.007) |
| Lev | 0.132*** | 0.131*** | 0.137*** | 0.138*** |
| | (10.281) | (10.201) | (10.707) | (10.751) |
| Revgrowth | −0.009* | −0.010* | −0.011** | −0.011** |
| | (−1.750) | (−1.907) | (−2.105) | (−2.088) |
| Div_ratio | −0.003 | −0.005 | −0.003 | −0.003 |
| | (−0.423) | (−0.701) | (−0.462) | (−0.513) |
| Taturn | −0.040*** | −0.038*** | −0.038*** | −0.038*** |
| | (−6.830) | (−6.378) | (−6.398) | (−6.362) |
| Indep_ratio | −0.012 | 0.011 | 0.011 | 0.010 |
| | (−0.330) | (0.282) | (0.300) | (0.264) |
| Top3_pay | 0.017*** | 0.016*** | 0.017*** | 0.016*** |
| | (5.063) | (4.850) | (4.910) | (4.844) |
| Year | controlled | controlled | controlled | controlled |
| Industry | controlled | controlled | controlled | controlled |
| Constant | 2.091*** | 2.210*** | 2.073*** | 2.192*** |
| | (37.315) | (39.701) | (37.014) | (39.521) |
| Observations | 23,973 | 23,973 | 23,973 | 23,973 |
| R-squared | 0.451 | 0.447 | 0.453 | 0.453 |
| F | 204.1 | 200.7 | 203.5 | 201.5 |

Note: t-statistics in parentheses, *** $p < 0.01$, ** $p < 0.05$, * $p < 0.1$.

increase in management compliance attention, the firm's ESG increases by an average of 0.041. This implies that when the firm's management more frequently mentions keywords such as "compliance", "supervision" and "risk management" in strategic discussions, its performance in environmental governance, social responsibility fulfillment, and corporate governance dimensions will significantly improve, thus validating H1. Board faultlines strength also exhibits a positive effect on ESG performance ($\beta = 0.344$, $p < 0.01$), indicating that for each unit increase in board faultlines strength, the firm's ESG rises by an average of 0.344. This suggests that the heterogeneity among board members in terms of gender, age, educational background, and other dimensions itself contributes to the improvement of ESG performance, validating H2. The interaction term between management compliance attention and board faultlines strength (Compliance Attention×Board Faultlines) has a coefficient of 0.114 ($p < 0.01$), which means that for each unit increase in board faultlines strength, the marginal effect of management compliance attention on ESG performance will increase by 0.114. This indicates that board faultlines strength significantly strengthens the promoting effect of management compliance attention on ESG performance, validating H3.

## 5. Robust checks

### 5.1. Marginal effect analysis

To further examine the tripartite relationship, we split the sample into high and low board faultlines strength groups (defined as mean ± 1 standard deviation of Board Faultlines) and analyze the effect of management compliance attention on ESG performance across these subgroups. The Fig 2 exhibits significant between-group differences: the slope of management compliance attention on ESG performance is steeper in the high-faultline group, indicating that stronger board faultlines amplify the compliance-ESG linkage. This visual confirmation reinforces the moderating role of board faultlines strength, providing additional support for H3.

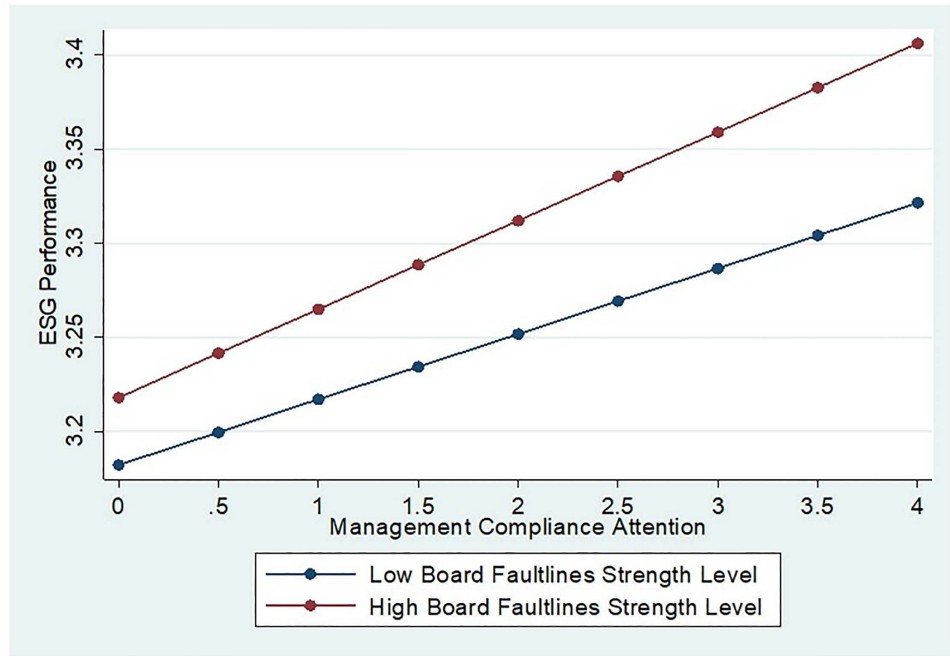

**Fig 2. Marginal effect analysis.**

## 5.2. Replacement of variables

To reinforce the research conclusions, this paper conducts robustness tests by substituting key variables in the model:

(1) Reconstructing the independent variable. We removed the keyword "law" from the original 11 compliance-related keywords to construct an adjusted management compliance attention variable (Compliance Attention$_{rb}$). The core function of law is to set a basic framework, while specific compliance practices require enterprises to extend it in combination with their own circumstances [54]. The primary purpose of excluding "law" is to test whether the impact of management compliance attention on ESG performance depends on the response to a single mandatory pressure. If the core relationship remains significant after excluding "law", it can more strongly prove that the impact of management compliance attention on ESG performance stems from the systematic cognition and response to the institutional environment, rather than merely relying on the passive mention of legal provisions. The behavior of decision-makers depends on the issues and answers they focus on, and is contingent on specific contexts [55]. As a strongly symbolic keyword, "law" may limit management's attention to compliance to the specific cognitive framework of "legal provisions", ignoring broader compliance dimensions (such as implementation mechanisms and emerging fields). Whether management compliance attention, after removing "law", still positively affects ESG performance will directly verify the driving role of "the cognitive breadth and depth of management compliance attention" in ESG practices.

This substitution was designed to explicitly test the robustness of our core findings against the exclusion of a foundational legal compliance element. As a cornerstone of regulatory frameworks, "law" inherently represents mandatory institutional pressures and forms the basis of corporate compliance systems. By excluding this pivotal keyword, we sought to verify whether the positive impact of management compliance attention on ESG performance is contingent on legal terminology or reflects a more holistic compliance orientation. Model 1 of Table 5 shows that the coefficient of Compliance Attention$_{rb}$ remains significantly positive ($\beta = 0.041$, $p < 0.01$), the Board Faultlines coefficient is still significantly positive ($\beta = 0.345$, $p < 0.01$), and the interaction term also retains significantly positive ($\beta = 0.117$, $p < 0.01$). These results are not significantly different from those of the baseline regression, indicating that the promoting effect of management compliance attention on ESG performance does not stem from passive responses to legal provisions, but rather reflects systematic attention to multi-dimensional compliance requirements such as "compliance foundation", "implementation mechanisms" and "emerging fields".

(2) Reconstructing the moderating variable. The previous text drew on the theoretical framework of group faultlines [11] and construction ideas [50]. Based on the demographic characteristics of board members (gender, age, education, and tenure), IA (internal group alignment index) and CGAI (cross-group alignment index) were calculated, forming the board faultlines strength formula IA × (1-CGAI). Due to the different specific calculation methods of the faultlines strength formula, two board faultlines strength variables, Board Faultlines (FLS) and NBoard Faultlines (NFLS), were formed. In the baseline regression model, the variable "Board Faultlines" was used. In the robustness test, variable replacement can be considered using another method to construct variables to replace the original variables. Therefore, anothor board faultlines strength indicator (NBoard Faultlines) is used instead of the original variable "Board Faultlines". Model 2 of Table 5 shows that the Compliance Attention coefficient remains 0.041 ($p < 0.01$), the NBoard Faultlines coefficient is 0.342 ($p < 0.01$), and the interaction term coefficient is 0.129 ($p < 0.01$), maintaining statistical significance at the 1% level respectively.

(3) Replacing both independent and moderating variables. Model 3 of Table 5 presents results when using both the adjusted independent variable (Compliance Attention$_{rb}$) and the new moderating variable (NBoard Faultlines). The Compliance Attention$_{rb}$ coefficient is 0.041 ($p < 0.01$), NBoard Faultlines coefficient is 0.342 ($p < 0.01$), and the interaction term coefficient is 0.132 ($p < 0.01$), with all key parameters retaining significance at the 1% threshold.

**Table 5. Replacement of variables.**

| VARIABLES | Model 1 ESG | Model 2 ESG | Model 3 ESG |
|---|---|---|---|
| Compliance Attentionrb | 0.041*** | | 0.041*** |
| | (15.893) | | (15.876) |
| Board Faultlines | 0.345*** | | |
| | (8.937) | | |
| Compliance Attentionrb × Board Faultlines | 0.117*** | | |
| | (2.975) | | |
| Compliance Attention | | 0.041*** | |
| | | (15.986) | |
| NBoard Faultlines | | 0.342*** | 0.342*** |
| | | (9.072) | (9.089) |
| Compliance Attention × NBoard Faultlines | | 0.129*** | |
| | | (3.373) | |
| Compliance Attentionrb × NBoard Faultlines | | | 0.132*** |
| | | | (3.486) |
| Lev | 0.139*** | 0.139*** | 0.140*** |
| | (10.825) | (10.826) | (10.902) |
| Revgrowth | −0.011** | −0.011** | −0.011** |
| | (−2.063) | (−2.116) | (−2.091) |
| Div_ratio | −0.003 | −0.004 | −0.004 |
| | (−0.511) | (−0.538) | (−0.536) |
| Taturn | −0.037*** | −0.037*** | −0.037*** |
| | (−6.302) | (−6.341) | (−6.280) |
| Indep_ratio | 0.010 | 0.012 | 0.011 |
| | (0.262) | (0.312) | (0.308) |
| Top3_pay | 0.016*** | 0.016*** | 0.016*** |
| | (4.864) | (4.801) | (4.822) |
| Year | controlled | controlled | controlled |
| Industry | controlled | controlled | controlled |
| Constant | 2.187*** | 2.191*** | 2.186*** |
| | (39.420) | (39.520) | (39.418) |
| Observations | 23,973 | 23,973 | 23,973 |
| R-squared | 0.453 | 0.453 | 0.453 |
| F | 201.5 | 201.7 | 201.6 |

Note: t-statistics in parentheses, *** $p < 0.01$, ** $p < 0.05$, * $p < 0.1$.

Collectively, these robustness test results consistently validate Hypotheses H1, H2, and H3, demonstrating the stability of our core findings across alternative variable operationalizations.

### 5.3. Endogenous issue

Based on the benchmark regression results, management compliance attention effectively promotes ESG performance; however, improved ESG performance might in turn motivate boards to prioritize corporate compliance, creating potential endogeneity arising from reverse causality. To address this concern, Model 1 of Table 6 presents results from a lagged

 

**Table 6. Endogenous issue.**

| VARIABLES | Model 1 one year lagged | 2SLS based on one year lagged instrumental variable | |
| --- | --- | --- | --- |
| | | Model 2 first stage | Model 3 second stage |
| | ESG | Compliance Attention | ESG |
| I.Compliance Attention | 0.035*** | | |
| | (12.207) | | |
| I.Board Faultlines | 0.301*** | | |
| | (6.732) | | |
| I.Compliance Attention×I.Board Faultlines | 0.111** | | |
| | (2.518) | | |
| Compliance Attentionpr1 | | | 0.057*** |
| | | | (12.183) |
| Board Faultlines | | | 0.306*** |
| | | | (7.078) |
| Compliance Attentionpr1×Board Faultlines | | | 0.177*** |
| | | | (2.951) |
| I.Compliance Attention | | 0.614*** | |
| | | (101.367) | |
| I.Employee | 0.035*** | 0.011** | |
| | (14.430) | (2.201) | |
| I.Lev | 0.111*** | −0.100*** | |
| | (7.692) | (−3.308) | |
| I.Div_ratio | 0.011 | −0.011 | |
| | (1.489) | (−0.680) | |
| I.Revgrowth | −0.000 | 0.000 | |
| | (−0.655) | (0.620) | |
| I.Taturn | −0.031*** | 0.000 | |
| | (−4.649) | (0.011) | |
| I.Indep_ratio | −0.008 | −0.039 | |
| | (−0.187) | (−0.449) | |
| I.Top3_pay | 0.017*** | −0.004 | |
| | (4.491) | (−0.540) | |
| Employee | | | 0.034*** |
| | | | (14.244) |
| Lev | | | 0.130*** |
| | | | (8.826) |
| Revgrowth | | | −0.004 |
| | | | (−0.608) |
| Div_ratio | | | 0.002 |
| | | | (0.310) |
| Taturn | | | −0.033*** |
| | | | (−4.882) |
| Indep_ratio | | | 0.002 |
| | | | (0.039) |
| Top3_pay | | | 0.017*** |
| | | | (4.395) |

*(Continued)*

**Table 6.** (Continued)

| | Model 1 one year lagged | 2SLS based on one year lagged instrumental variable | |
| --- | --- | --- | --- |
| | | Model 2 first stage | Model 3 second stage |
| Year | controlled | controlled | controlled |
| Industry | controlled | controlled | controlled |
| Constant | 2.224*** | 1.534*** | 2.173*** |
| | (34.622) | (11.240) | (33.459) |
| Observations | 17,353 | 17,353 | 17,353 |
| R-squared | 0.471 | 0.587 | 0.472 |
| F | 160.1 | 261.3 | 160.9 |

Note: t-statistics in parentheses,*** $p < 0.01$, ** $p < 0.05$, * $p < 0.1$.

regression where independent, moderating, and control variables (excluding year and industry) are lagged by one period. The coefficient for lagged compliance attention (l.Compliance Attention) is 0.035 ($p < 0.01$), supporting H1. The lagged board faultlines strength (l.Board Faultlines) coefficient is 0.301 ($p < 0.01$), validating H2, and the interaction term coefficient is 0.111 ($p < 0.05$), confirming H3.

Model 2 and Model 3 employ the Instrumental Variable (IV) method to further mitigate endogeneity, using one-period lagged management compliance attention as the instrumental variable and applying two-stage least squares (2SLS) regression. The one-period lagged variable occurs prior to the current dependent variable; theoretically, it can influence the current dependent variable indirectly by affecting the current explanatory variable, rather than being directly impacted by the reverse influence of the current dependent variable. This temporal precedence significantly reduces the endogeneity risk caused by bidirectional causality, thus qualifying it as an effective instrumental variable for the current explanatory variable. The results of Model 2 show that the coefficient of l.Compliance Attention is 0.614 ($p < 0.01$), with an R-squared of 0.587, indicating a high goodness-of-fit of the model and a strong correlation between the instrumental variable and the endogenous variable, which satisfies the relevance condition. The fitting variable (Compliance Attention_pr1) of Model 2 regression is introduced into Model 3. In Model 3, the coefficient of compliance attention (Compliance Attention_pr1) is 0.057 ($p < 0.01$), the coefficient of board faultlines strength (Board Faultlines) is 0.306 ($p < 0.01$), and the interaction term coefficient is 0.177 ($p < 0.01$), suggesting that after accounting for reverse endogeneity, the regression results remain consistent with previous hypotheses. These findings strengthen the robustness of the core conclusion regarding the issue of causal direction.

### 5.4. Heterogeneity analysis

Companies may exhibit heterogeneous compliance attention levels based on firm characteristics, which in turn translate into varying levels of compliance-building efforts and distinct ESG performance outcomes. To explore this, we conduct grouping tests based on market competition intensity and ownership structure. Models 1–2 of Table 7 present results for high- and low-competition groups respectively. In a high-competition environment (Model 1), the coefficient of management compliance attention (Compliance Attention) on ESG performance is 0.039 ($p < 0.01$); whereas in a low-competition environment (Model 2), this coefficient rises to 0.047 ($p < 0.01$). This indicates that enterprises in low-competition environments achieve more significant improvements in ESG performance from compliance attention. For each unit increase in compliance attention, the ESG performance of enterprises in low-competition environments increases by 0.008 more than that of enterprises in high-competition environments, likely because reduced competitive pressures allow more resources

**Table 7. Groups analyses.**

| VARIABLES | Model 1 | Model 2 | Model 3 | Model 4 |
|---|---|---|---|---|
| | High competition | Low competition | State-owned companies | Private companies |
| | ESG | ESG | ESG | ESG |
| Compliance Attention | 0.039*** | 0.047*** | 0.045*** | 0.037*** |
| | (13.352) | (8.898) | (10.432) | (11.553) |
| Board Faultlines | 0.338*** | 0.349*** | 0.391*** | 0.306*** |
| | (7.544) | (4.606) | (6.538) | (5.889) |
| Compliance Attention × Board Faultlines | 0.078* | 0.192** | 0.087 | 0.112** |
| | (1.694) | (2.509) | (1.469) | (2.076) |
| Employee | 0.035*** | 0.051*** | 0.040*** | 0.033*** |
| | (13.860) | (12.804) | (11.908) | (11.601) |
| Lev | 0.139*** | 0.141*** | 0.096*** | 0.165*** |
| | (9.294) | (5.660) | (4.479) | (10.096) |
| Revgrowth | −0.010 | −0.011 | −0.004 | −0.009 |
| | (−1.589) | (−1.096) | (−0.452) | (−1.306) |
| Div_ratio | −0.002 | −0.010 | −0.002 | −0.003 |
| | (−0.240) | (−0.797) | (−0.156) | (−0.396) |
| Taturn | −0.031*** | −0.048*** | −0.043*** | −0.036*** |
| | (−4.263) | (−4.607) | (−4.682) | (−4.571) |
| Indep_ratio | −0.062 | 0.180** | −0.021 | 0.003 |
| | (−1.443) | (2.443) | (−0.353) | (0.066) |
| Top3_pay | 0.022*** | 0.003 | 0.015*** | 0.016*** |
| | (5.594) | (0.473) | (2.634) | (3.836) |
| Year | controlled | controlled | controlled | controlled |
| Industry | controlled | controlled | controlled | controlled |
| Constant | 2.720*** | 2.199*** | 2.202*** | 2.273*** |
| | (43.763) | (23.660) | (25.144) | (30.865) |
| Observations | 17,464 | 6,509 | 9,187 | 14,786 |
| R-squared | 0.457 | 0.452 | 0.452 | 0.468 |
| F | 266.3 | 63.05 | 77.27 | 138.7 |

Note: t-statistics in parentheses,*** $p < 0.01$, ** $p < 0.05$, * $p < 0.1$.

to be channeled into sustained compliance efforts. Analysis of moderating effects reveals that the interaction term coefficient in the low-competition group (Model 2) is 0.192 ($p < 0.05$), which is higher than 0.078 ($p < 0.1$) in the high-competition group (Model 1). This suggests that strong board faultlines are particularly effective in amplifying the compliance-ESG link when market competition is low, enabling boards to better leverage diverse perspectives for compliance-driven sustainability improvements.

On the ownership dimension, Model 3 (state-owned companies) and Model 4 (private companies) of Table 7 show that the Compliance Attention coefficient is 0.045 ($p < 0.01$) in the state-owned companies group, which is higher than 0.037 ($p < 0.01$) in the private companies group. This aligns with post-2008 policy mandates targeting state-owned companies, which impose institutional pressures to prioritize compliance and sustainability. Notably, the interaction term between Compliance Attention and Board Faultlines loses significance in state-owned companies, whereas it remains significant ($p < 0.05$) at 0.112 in private companies. A plausible explanation is that state-owned companies face strong external policy

and internal governance pressures that directly drive ESG performance, making board faultlines less critical for enhancing compliance attention. In contrast, private companies—often characterized by more diverse board compositions and governance structures—benefit from faultline-driven diversity, which enhances managerial compliance awareness and synergistically boosts ESG performance through improved strategic oversight and stakeholder engagement.

## 5.5. The mediating effect of organizational resilience

Organizational resilience—a critical dynamic capability—focuses on agile processing of external shocks and strategic responses. Drawing on dynamic capabilities theory [54], firms must integrate and reconfigure internal/external resources to cultivate adaptive capabilities that enable navigation of volatile market environments. Management compliance attention drives companies to establish systematic compliance systems by responding to mandatory and normative pressures, transforming external pressures into momentum for building internal resilience. Firms reduce legal risks by establishing rigid compliance with regulatory requirements [35], and laying a "bottom-line security" foundation for organizational resilience. Compliance practices directly enhance companies' early warning and response capabilities to financial and operational risks [56], and firms that invest in data governance resources in advance can more quickly adapt to upgraded data security regulations [57], reflecting the dynamic adaptation dimension of resilience. This "proactive compliance-capability building" model aligns with the internal logic of dynamic capabilities theory [58] regarding "resource integration to address environmental changes".

Organizational resilience significantly contributes to improving ESG performance. From an environmental (E) perspective, companies with strong organizational resilience can maintain long-term green technology investments in the face of environmental goals such as "dual carbon" by virtue of their risk resistance and stable decision-making processes, without abandoning environmental projects due to short-term cost pressures [27]. In the social (S) dimension, companies reduce risks such as labor disputes and data breaches through sound compliance systems and strengthened supply chain compliance management, maintaining good trust relationships with stakeholders such as employees and partners [59]. At the governance (G) level, boards of directors strengthen internal checks and balances through mechanisms such as establishing independent audit committees, directly improving the transparency of corporate governance [60]. Moreover, organizational resilience has a dual effect of risk buffering and value creation: risk management processes effectively identify and reduce uncertainties in ESG practices, mitigating the negative impact of regulatory penalties on ESG scores [61].

Adopting the mediating effect framework proposed by Wen & Ye (2014) [62], we specify the following models to test the hypothesized mediating role of organizational resilience.

$$
\begin{aligned}
ESG_{it} = {} & \beta_0 + \beta_1 Compliance\ Attention_{it} + \beta_2 Employee_{it} + \beta_3 Lev_{it} \\
& + \beta_4 Revgrowth_{it} + \beta_5 Div\_ratio_{it} + \beta_6 Taturn_{it} \\
& + \beta_7 Indep\_ratio_{it} + \beta_8 Top3\_pay_{it} \\
& + \sum Industry + \sum Year + \varepsilon_{it}
\end{aligned}
\tag{11}
$$

$$
\begin{aligned}
Orgre_{it} = {} & \beta_0 + \beta_1 Compliance\ Attention_{it} + \beta_2 Employee_{it} + \beta_3 Lev_{it} \\
& + \beta_4 Revgrowth_{it} + \beta_5 Div\_ratio_{it} + \beta_6 Taturn_{it} \\
& + \beta_7 Indep\_ratio_{it} + \beta_8 Top3\_pay_{it} \\
& + \sum Industry + \sum Year + \varepsilon_{it}
\end{aligned}
\tag{12}
$$

$$
\begin{aligned}
ESG_{it} = {} & \beta_0 + \beta_1 Compliance\ Attention_{it} + \beta_2 Orgre_{it} + \beta_3 Employee_{it} \\
& + \beta_4 Lev_{it} + \beta_5 Revgrowth_{it} + \beta_6 Div\_ratio_{it} + \beta_7 Taturn_{it} \\
& + \beta_8 Indep\_ratio_{it} + \beta_9 Top3\_pay_{it} + \sum Industry + \sum Year + \varepsilon_{it}
\end{aligned}
\tag{13}
$$

We obtained the mediating variable (Orgre) of organizational resilience through the following steps: (1) Calculate the cumulative sales revenue growth over a three-year period; (2) Calculate the standard deviation of monthly stock returns over a one-year period; (3) Use the entropy weight method to calculate the comprehensive indicator for the above two standardized indicators.

Table 8 reports the mediating effect test results. Compliance Attention in Model 1 significantly and positively impacts Orgre ($\beta=0.00077$, $p<0.01$), indicating that for each unit increase in management compliance attention, organizational resilience increases by an average of 0.00077. Although this value seems small, it reflects the role of management compliance attention in enhancing organizational adaptability by establishing systematic risk response mechanisms. The significant positive effect ($\beta=0.06527$, $p<0.01$) of Compliance Attention on ESG performance in Model 2 indicates that management compliance attention translates into the effectiveness of ESG practices. After introducing the mediating variable of Orgre in Model 3, the coefficient of Orgre is significantly positive ($\beta=0.93657$, $p<0.01$), and the coefficient of Compliance Attention remains significantly positive ($\beta=0.06455$, $p<0.01$), indicating that each unit increase in organizational resilience raises the natural logarithm of ESG scores by an average of 0.94, reflecting that enterprise dynamic adaptability significantly improves ESG performance. The coefficient of management compliance attention decreases to 0.06455 ($p<0.01$), which is still significantly positive but slightly lower than 0.06527 in Model 2, indicating that organizational

**Table 8. Mediating effect.**

| VARIABLES | Model 1 Orgre | Model 2 ESG | Model 3 ESG |
|---|---|---|---|
| Orgre | | | 0.93657*** |
| | | | (11.49330) |
| Compliance Attention | 0.00077*** | 0.06527*** | 0.06455*** |
| | (3.20751) | (21.54847) | (21.36452) |
| Employee | 0.00433*** | 0.03769*** | 0.03363*** |
| | (22.60578) | (15.55229) | (13.76923) |
| Lev | −0.00558*** | 0.00550 | 0.01073 |
| | (−4.72052) | (0.36843) | (0.71979) |
| Revgrowth | −0.00829*** | −0.02463*** | −0.01687*** |
| | (−16.32217) | (−3.83622) | (−2.61960) |
| Div_ratio | 0.00308*** | 0.00898 | 0.00609 |
| | (4.81016) | (1.10852) | (0.75384) |
| Taturn | −0.00018 | 0.02220*** | 0.02237*** |
| | (−0.33633) | (3.34995) | (3.38409) |
| Indep_ratio | −0.00098 | −0.02906 | −0.02814 |
| | (−0.27994) | (−0.65830) | (−0.63931) |
| Top3_pay | 0.00011 | 0.00807** | 0.00797** |
| | (0.34219) | (2.02715) | (2.00727) |
| Year | controlled | controlled | controlled |
| Industry | controlled | controlled | controlled |
| Constant | 0.80411*** | 2.27681*** | 1.52370*** |
| | (38.89917) | (8.71272) | (5.67025) |
| Observations | 23,973 | 23,973 | 23,973 |
| R-squared | 0.73389 | 0.20129 | 0.20567 |
| F | 1784 | 163.0 | 163.1 |

Note: t-statistics in parentheses,*** $p<0.01$, ** $p<0.05$, * $p<0.1$.

resilience plays a partial mediating role between compliance attention and ESG performance. Both the bootstrap test (117 resamplings, Z = 3.24, p = 0.001, 95% CI (0.0003, 0.0012)) and the Sobel test (Z = 3.089, p = 0.002) indicate a significant mediating effect of organizational resilience. In statistics, when |Z| ≥ 2.58, the mediating effect is significant at the 1% level. Given that the Z-value (3.24) of the bootstrap test with 117 resamplings is greater than 2.58, this indicates that the mediating effect of organizational resilience is significant at the 1% level. Meanwhile, the 95% confidence interval (0.0003, 0.0012) does not include zero, further confirming that the mediating effect of organizational resilience is significant. Considering further that the Sobel test Z-value (3.089) exceeds 2.58, the mediating effect of organizational resilience in the compliance-ESG link still remains significant at the 1% level. Compliance attention not only enhances ESG perception through direct signal transmission but also achieves sustainable performance improvement by shaping organizational resilience.

## 6. Conclusions

This study examines the impact of management compliance attention on ESG performance using a sample of Chinese A-share listed companies from 2010 to 2022, exploring the moderating role of board faultlines strength and the mediating role of organizational resilience. Based on the empirical analysis, the following main conclusions are drawn.

First, management compliance attention drives ESG performance. Enterprises gain legitimacy by adapting to external regulations and social expectations; management focus on compliance reflects active alignment with institutional elements like regulatory requirements and social norms. This positive association demonstrates how institutional mechanisms operate: strong compliance attention systematically promotes sustainable practices in environmental (e.g., ISO 14001 adoption), social (e.g., labor standard compliance), and governance (e.g., independent director ratios) domains. Management compliance attention reflects its "compliance-ESG linkage" cognitive framework. Internalizing external institutional pressures into cognitive attention guides enterprises to integrate compliance requirements with ESG practices, promoting performance improvements and verifying cognitive theory. Institutional and cognitive theories together form a logic where external institutional constraints and internal cognitive transformation drive ESG performance improvements.

Second, board faultlines strength positively impacts ESG performance, and its interaction with management compliance attention strengthens compliance attention's promoting effect on ESG performance. This indicates diverse board compositions—varying in gender, age, education, and professional experience—bring diverse cognitive frameworks to compliance and ESG discussions. Diverse board perspectives enable richer discussions, helping supervise management in translating compliance attention into specific ESG actions. The verified moderating role of board faultlines strength aligns with upper echelons theory core logic: board heterogeneity affects depth and breadth of ESG-related strategic decisions, influencing enterprise practices.

Third, organizational resilience partially mediates between management compliance attention and ESG performance. This finding enriches dynamic capabilities theory, showing compliance attention enhances enterprises' ability to adapt to environmental changes and allocate resources strategically. Management compliance attention essentially constructs systematic resource allocation and risk response mechanisms, enhancing adaptability to policy and market changes. This ability strengthens organizational resilience and ensures sustainability and effectiveness of ESG practices. The verified mediating role of organizational resilience expands dynamic capabilities theory application in ESG, revealing compliance attention drives dynamic capability cultivation and promotes continuous ESG performance improvements by enhancing resilience.

Fourth, heterogeneity analysis shows effects vary by ownership structure and market competition intensity. State-owned enterprises exhibit stronger compliance-driven ESG performance. From an institutional theory perspective, state-owned enterprises embed more deeply in institutional environments, with ESG practices closely aligned to policy orientations, making management compliance attention more likely to translate into improved ESG performance. From an upper echelons theory perspective, state-owned enterprise management often has stronger policy implementation

awareness, with cognitive frameworks more closely linking compliance requirements to ESG-related strategic goals, promoting more active resource allocation to ESG practices. Regarding market competition intensity, management compliance attention impacts ESG more significantly in low-competition environments, indicating enterprises rely more on internal governance to promote ESG practices amid weaker market competition.

## 7. Theoretical implications

This study contributes to theoretical streams by unpacking the mechanisms through which management compliance attention and board faultlines strength shape ESG performance, thereby enriching our understanding of corporate sustainability.

### 7.1. Integrating institutional theory and cognitive theory through compliance as a dual-driven legitimacy mechanism

Institutional theory holds that organizations gain legitimacy by aligning with external regulative, normative, and cognitive institutions. Cognitive theory emphasizes how management translates abstract institutional pressures into actionable strategies through interpretation and meaning attribution. This study integrates these perspectives, revealing management compliance attention as a critical interface connecting institutional requirements and ESG performance, deepening theoretical understanding of sustainable development drivers. Institutional theory alone explains why organizations respond to external pressures but not how these pressures undergo internal processing. Management interprets institutional signals through cognitive frameworks, converting institutional pressures into targeted ESG practices. This dual-driven mechanism expands institutional theory by highlighting cognitive agency in institutional adaptation and enriches cognitive theory by situating management attention within institutional constraints.

### 7.2. Extending upper echelons theory through board faultlines as structural drivers of ESG

Upper echelons theory emphasizes how top management characteristics influence organizational outcomes, but this study shifts focus to board faultlines as a deeper structural dimension of diversity. Board faultlines strength positively moderates management compliance attention's effect on ESG performance. This shows demographic and experiential differences among directors amplify strategic oversight and innovation. This challenges viewing board diversity as a mere demographic checklist. It highlights faultline dynamics' importance in shaping ESG outcomes.It contributes to upper echelons theory by showing that faultlines create cognitive diversity that fosters critical dialogue on sustainability, thereby enhancing ESG strategy formulation and implementation.

### 7.3. Enriching dynamic capabilities theory through resilience as a mediating mechanism

Dynamic capabilities theory emphasizes firms' ability to adapt and reconfigure resources. Specific pathways through which compliance enhances such capabilities remain understudied. Compliance attention strengthens resilience. This finding provides empirical evidence for how regulatory focus builds foundational capabilities underpinning sustainable practices. Integrating compliance into the dynamic capabilities framework, the research shows compliance is not a standalone activity. It is a capability-enhancing process enabling firms to align ESG goals with operational resilience. This extends the theory to include regulatory compliance as a driver of adaptive capacity.

### 7.4. Contributing to transitional economy research through institutional context variations

Focusing on Chinese A-share listed companies, this study reveals how ownership structure and market competition moderate compliance effects under varying competition degrees and ownership institutions. State-owned companies leverage policy pressures to drive ESG performance. Private companies rely on board diversity to compensate for

insufficient institutional mandates. This highlights institutional environments' differentiated roles in shaping compliance behaviors. It contributes to transitional economy research by demonstrating formal (state policies) and informal (board governance) mechanisms interact to influence sustainability outcomes. It underscores the need to contextualize compliance theories within institutional settings with varying regulatory stringency and corporate governance practices.

## 8. Practical implications

The findings of this study offer actionable insights for firms, policymakers, and stakeholders seeking to enhance ESG performance through strategic compliance management and governance reforms. Below are tailored practical implications derived from the research.

### 8.1. Integrating compliance attention and board faultlines into strategic ESG frameworks

Companies should embed management compliance attention into core strategies by prioritizing regulatory and ethical keywords in managerial discourse, establishing dedicated compliance governance mechanisms, and aligning compliance reporting with ESG goals. Firms should cultivate boards with diverse demographic and experiential backgrounds to strengthen faultlines, enabling richer discussions and more robust ESG oversight. By intentionally designing boards with varied expertise in areas like sustainability, governance, and social responsibility, firms can harness faultline dynamics to identify ESG risks, and develop innovative solutions.

### 8.2. Building resilience through compliance-ESG synergy

Firms should link compliance systems to organizational resilience by integrating regulatory adherence with adaptive capabilities such as scenario planning, flexible resource allocation, and risk management frameworks. Compliance audits and risk assessments can serve as tools to identify ESG gaps, guiding investments in resilience-building initiatives like green technology adoption, employee upskilling, or supply chain diversification. This synergy ensures compliance efforts enhance the firm's ability to navigate external shocks while advancing ESG performance.

### 8.3. Aligning compliance and ESG with stakeholder expectations

From the stakeholder perspective, firms should integrate management compliance attention with ESG performance by systematically addressing the expectations of diverse stakeholders. For shareholders, enhancing ESG disclosure to highlight compliance-driven risk mitigation and long-term value creation can safeguard interests and attract ESG-focused investment. For employees, strengthening labor compliance fosters trust and psychological ownership, while leveraging board faultlines ensures employee rights are embedded in ESG strategies. For customers and suppliers, extending compliance requirements to supply chains creates a compliance ecosystem, enhancing product sustainability and meeting ethical consumption demands. For regulators and communities, proactive alignment with policies and public disclosure of compliance achievements secure regulatory favor and social legitimacy. By institutionalizing stakeholder feedback mechanisms, firms can translate multi-stakeholder needs into actionable compliance measures, thereby enhancing ESG performance and fostering collaborative sustainability.

### 8.4. Tailoring strategies to ownership and competitive contexts

State-owned companies should leverage policy mandates to align compliance attention with national sustainability goals, using their institutional resources to lead in ESG areas like renewable energy and social equity. Private firms, lacking strong external compliance pressures, should prioritize board diversity to amplify compliance effects, recruiting directors with varied backgrounds to enhance ESG oversight and integrate compliance into core strategies. In low-competition

industries, proactive compliance and ESG investments can differentiate firms, build reputational capital, and prepare for evolving regulatory demands.

## 9. Limitations and future research

### 9.1. Study limitations

This study, while contributing to the literature on compliance, governance, and ESG performance, has several limitations that warrant acknowledgment. First, the research is contextualized within Chinese A-share listed companies from 2010 to 2022, which may limit generalizability to non-listed firms, non-Chinese contexts, or periods outside the sample timeframe. Institutional environments, regulatory pressures, and board dynamics vary significantly across regions and firm types, potentially affecting the robustness of the findings in different settings. Second, the operationalization of management compliance attention relies on keyword frequency in annual reports, which may not fully capture the depth of compliance implementation or managerial cognition. Textual analysis, while objective, could overlook implicit compliance practices or the quality of compliance efforts. Third, while heterogeneity analysis considers ownership structure and market competition, the study does not delve into industry-specific effects (e.g., high-regulation sectors like finance or manufacturing) or regional disparities within China, which could offer additional insights into contextual contingencies.

### 9.2. Directions for future research

To address these limitations, future studies could pursue several avenues. First, expanding the sample to include non-listed firms, cross-national comparisons, or longitudinal designs would enhance the generalizability of the findings and shed light on how institutional contexts shape the compliance-ESG relationship. Second, triangulating management compliance attention with qualitative data (e.g., management interviews) or behavioral indicators (e.g., regulatory violation records) would improve measurement validity and capture both symbolic and substantive compliance. Third, investigating industry-specific dynamics, such as how compliance attention affects ESG performance in high-pollution industries versus technology sectors, could provide tailored insights for regulatory design and corporate strategy.

## Supporting information

**S1 File. DTA do excel.**
(ZIP)

## Author contributions

**Conceptualization:** Yong Jiang, Fei Han.

**Data curation:** Yong Jiang.

**Methodology:** Yong Jiang.

**Project administration:** Yong Jiang, Fei Han.

**Supervision:** Yong Jiang, Fei Han.

**Writing – original draft:** Fei Han.

**Writing – review & editing:** Yong Jiang, Fei Han.

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
