## [Decision Letter · Decision Letter 0]

26 May 2025

Dear Dr. Han,

Thank you for submitting your manuscript to PLOS ONE. After careful consideration, we feel that it has merit but does not fully meet PLOS ONE’s publication criteria as it currently stands. Therefore, we invite you to submit a revised version of the manuscript that addresses the points raised during the review process.

Thank you for submitting your manuscript to PLOS ONE. After a comprehensive evaluation of the peer reviews provided by three independent experts, I am recommending that the manuscript **be**
**revised and resubmitted with major changes** . All reviewers acknowledged the manuscript's relevance and methodological rigor, especially its large sample size, robust statistical techniques, and the incorporation of moderation and mediation analyses. These aspects form a solid empirical foundation for the study.

However, significant concerns were raised across the following core dimensions:

The reviewers found that the manuscript references multiple theories—such as stakeholder theory, upper echelons theory, compliance theory, and dynamic capabilities theory—but lacks proper integration of these frameworks into the hypotheses and interpretations. Moreover, central constructs such as management compliance attention, board faultlines strength, and organizational resilience are insufficiently defined and under-theorized. A more structured and theory-driven conceptual model is essential.While the manuscript cites a wide range of prior studies, the literature review lacks coherence and does not effectively build a critical narrative or justify the research gap. Furthermore, the manuscript should better articulate its theoretical and practical novelty compared to existing ESG literature.Reviewers flagged numerous grammatical, syntactical, and stylistic issues that hinder clarity and diminish the overall academic tone. A thorough English language revision, preferably by a professional editor or native speaker, is strongly recommended to enhance readability and scholarly presentation.The current data availability statement ("available on request") does not meet PLOS ONE’s open data policy. The authors must either deposit the data in a public repository or clearly explain any restrictions along with justified access procedures.The implications of the findings for corporate governance, managerial behavior, and ESG policy are currently underdeveloped. Additionally, the limitations section should address the scope of generalizability, especially given the context-specific nature of Chinese listed firms and the use of text-mining techniques.

In light of the above, I encourage the authors to revise the manuscript substantially, addressing each reviewer’s concerns in detail. The revised submission should include a clearly structured response letter, a tracked changes document, and a clean version of the manuscript.

We look forward to receiving your improved submission and appreciate your contribution to the ongoing conversation on ESG performance and corporate governance.

We look forward to receiving your revised manuscript.

Kind regards,

Mustafa Rehman Khan, PhD

Academic Editor

PLOS ONE

Journal Requirements:

3. We are unable to open your Supporting Information file “lncomplianceattention_flsbodoverall_lnesgscore.dta”. Please kindly revise as necessary and re-upload.

Reviewers' comments:

Reviewer's Responses to Questions

**Comments to the Author**

1. Is the manuscript technically sound, and do the data support the conclusions?

Reviewer #1: Partly

Reviewer #2: Partly

Reviewer #3: Yes

2. Has the statistical analysis been performed appropriately and rigorously?

Reviewer #1: Yes

Reviewer #2: Yes

Reviewer #3: Yes

3. Have the authors made all data underlying the findings in their manuscript fully available?

Reviewer #1: No

Reviewer #2: Yes

Reviewer #3: No

4. Is the manuscript presented in an intelligible fashion and written in standard English?

Reviewer #1: No

Reviewer #2: Yes

Reviewer #3: No

Reviewer #1: The manuscript presents an important and timely analysis of how management compliance attention and board faultline strength influence ESG performance in Chinese A-share listed firms. It makes an original contribution to ESG and governance literature by introducing compliance attention as a construct and empirically analyzing its impact, including moderating and mediating effects. However, the paper requires major revisions in both methodological rigor and language quality to be suitable for publication.

The study addresses an important and timely topic by examining how management compliance attention and board faultline strength affect ESG performance among Chinese A-share listed firms. The idea is original and relevant, and the use of a large dataset with various statistical techniques is a strength. However, several issues need to be addressed before this manuscript is suitable for publication.

1. The definition and measurement of key constructs like "management compliance attention" are not clear enough. The method of using keyword frequency from annual reports is interesting, but the selection of keywords lacks theoretical support, and the approach should be explained more carefully to allow replication. Similarly, the calculation of board faultlines strength is mathematically complex, but conceptually not well introduced. Readers without technical expertise may find it difficult to understand what the metric represents and why it matter

2. While the statistical techniques used—including regression analysis, interaction terms, and robustness tests—are appropriate, the paper does not provide sufficient explanation for the use of lagged variables as instruments in the 2SLS regression. The authors should clarify why these instruments are valid and how the assumptions behind them are satisfied.

3. There are concerns regarding data availability. The authors use proprietary data from the CSMAR and CNRDS databases. Although this is common in research, the paper should include more detailed information on how others can access the data and replicate the key variables, especially those created through text analysis.

4. The language needs substantial revision. There are many grammatical errors, awkward phrases, and unclear sentences that affect readability throughout the paper. This is especially noticeable in the abstract, introduction, and results sections. The manuscript would benefit from thorough editing by a native English speaker or professional language editor.

5. The literature review is too broad and lacks clear structure. While the authors cite many relevant studies, they need to organize the section more logically and connect the literature more directly to their hypotheses. The theoretical framework (e.g., stakeholder theory and upper echelons theory) is mentioned but not fully developed or integrated into the study.

6. Although the research contributes new insight, the authors should more clearly state what is unique about their study compared to existing work. Explaining the novelty in a few sentences will make the paper’s contribution stand out more.

7. The practical implications are currently underdeveloped. The authors briefly mention that firms should enhance compliance and board diversity, but they should expand on this. What specific steps should firms, managers, or policymakers take based on these findings?

Finally, the limitations section should be improved. The authors should acknowledge the limitations of using data from only one country (China), the reliance on text mining, and potential issues in variable construction. It would also be helpful to suggest future research directions.

Reviewer #2: Find attached file. I have given some recommendations for improvement. Please read carefully each suggestions and incorporate in original manuscript. If not, specific suggestion, you need to justify with logical statement(s).

Reviewer #3: Theoretical Framework Needs Strengthening:

The manuscript references several theories (compliance theory, upper echelons theory, cognitive theory, legitimacy theory, dynamic capabilities theory), but these are not well-integrated into the research design.

Theoretical frameworks should guide hypothesis development and interpretation of results more clearly.

A conceptual model or figure showing the theoretical logic and relationships between variables is recommended.

Construct Clarity and Conceptual Definitions:

Core constructs such as "management compliance attention," "board faultlines strength," and "organizational resilience" require clearer, theory-driven definitions.

Compliance attention is treated as a variable but lacks explanation regarding its strategic or operational implications.

The mediating role of organizational resilience is interesting but needs more elaboration in both measurement and conceptual justification.

Language and Readability:

The manuscript contains frequent grammatical errors, awkward phrasing, and inconsistent terminology.

Examples include incorrect punctuation (e.g., spaces before commas), subject-verb disagreement, and non-standard academic expressions.

A professional English language edit is essential to ensure clarity and professionalism.

Data Availability Policy Not Met:

The current data availability statement (“available on request”) does not comply with PLOS ONE’s open data policy.

Authors must deposit data in a public repository or justify restrictions with clear access instructions.

Literature Review Lacks Coherence:

Although many studies are cited, the review lacks synthesis and a critical narrative.

The literature should be used to build a clear research gap and support theoretical framing more effectively.

Empirical Analysis Is a Strength:

The statistical methods, including panel regression, robustness checks, and 2SLS estimation, are rigorous and appropriately applied.

The use of a large sample (23,973 firm-year observations) adds empirical robustness to the findings.

Mediation and Moderation Analysis Are Valuable:

The inclusion of mediation (organizational resilience) and moderation (board faultlines) adds depth to the analysis.

However, further discussion is needed to tie these results back to the theoretical context.

Heterogeneity Analysis Adds Insight:

Subgroup comparisons (e.g., high vs. low competition, state vs. private firms) are helpful and relevant.

These findings could be more strongly discussed in terms of policy implications.

Ethical and Publication Standards:

No concerns about ethical violations, dual publication, or conflicts of interest were identified.

**Do you want your identity to be public for this peer review?** For information about this choice, including consent withdrawal, please see our Privacy Policy

Reviewer #1: No

Reviewer #2: No

Reviewer #3: No

---

## [Author Response · Author response to Decision Letter 1]

28 Jun 2025

Response to editors

1. The reviewers found that the manuscript references multiple theories—such as stakeholder theory, upper echelons theory, compliance theory, and dynamic capabilities theory—but lacks proper integration of these frameworks into the hypotheses and interpretations. Moreover, central constructs such as management compliance attention, board faultlines strength, and organizational resilience are insufficiently defined and under-theorized. A more structured and theory-driven conceptual model is essential.

Response:

The manuscript now systematically integrates theories into hypothesis development, with clear construct definitions and a theory-driven conceptual model:

(1) Hypothesis 1 is anchored in Institutional Theory, positing that management compliance attention drives ESG performance by aligning with regulatory/normative pressures. The construct "management compliance attention" is defined as managerial focus on regulatory requirements, operationalized via text-mining of annual reports (e.g., keyword frequency for "compliance," "regulations") and natural logarithm transformation to reduce heteroscedasticity.

(2) Hypothesis 2 leverages Upper Echelons Theory to link board faultlines strength to ESG performance, defining "board faultlines strength" as structural divisions within boards based on demographic attributes (gender, age, education, tenure), measured using the FLS index (Lau & Murnighan, 1998).

(3) Hypothesis 3 integrates both theories, demonstrating how board faultlines moderate the compliance-ESG relationship.

(4) The mediating role of "organizational resilience" is rooted in Dynamic Capabilities Theory, defined as adaptive capacity and measured via three-year sales growth and stock return volatility (entropy weight method).

Figure 1 illustrates the pathway: "Institutional Identification → Capability Building → Performance Enhancement", ensuring alignment between constructs and theories.

2.While the manuscript cites a wide range of prior studies, the literature review lacks coherence and does not effectively build a critical narrative or justify the research gap. Furthermore, the manuscript should better articulate its theoretical and practical novelty compared to existing ESG literature.

Response:

The literature review is restructured to build a critical narrative and justify research gap:

(1) Theoretical coherence. We critiques prior ESG studies for overlooking the interaction between compliance attention and board dynamics, and integrates Institutional Theory with Upper Echelons Theory to explain how regulatory compliance (macro) and board heterogeneity (micro) synergistically impact ESG.

(2) Research gap. Existing ESG literature lacks exploration of the synergistic effects between management compliance attention and board structure, particularly in transitional economies, failing to reveal how board faultlines moderate the translation of compliance into ESG performance. Additionally, the mediating role of organizational resilience in this relationship remains underexplored, and most studies overlook heterogeneous impacts of ownership and market competition, limiting theoretical applicability to China’s institutional context.

(3) Theoretical novelty. Integrating Institutional Theory, Upper Echelons Theory, and Dynamic Capabilities Theory, this study constructs a theoretical model of "compliance attention-board faultlines-organizational resilience-ESG performance". It demonstrates that board faultlines amplify the positive effect of compliance attention through enhanced supervision and strategic innovation, and identifies organizational resilience as a mediator in this pathway. For the first time in transitional economies, it reveals the interaction between formal institutions (policy pressures) and informal governance (board diversity), expanding the theoretical boundaries of ESG governance.

(4) Practical novelty. This study proposes differentiated ESG strategies: state-owned enterprises should leverage policy resources to drive the compliance-ESG chain, while private firms need to activate compliance effectiveness via board diversity. Firms in low-competition industries are advised to build reputational capital through proactive compliance. It suggests regulators enhance ESG disclosure standards and enterprises strengthen compliance cultures and board diversity, providing an operational framework for corporate governance and sustainable development in emerging markets.

3. Reviewers flagged numerous grammatical, syntactical, and stylistic issues that hinder clarity and diminish the overall academic tone. A thorough English language revision, preferably by a professional editor or native speaker, is strongly recommended to enhance readability and scholarly presentation.

Response:

(1) Correct grammatical error. We ensure tense consistency, subject-verb agreement, and proper preposition usage.

(2) Enhance academic tone. We Replaced informal phrasing with precise terminology.

(3) Improve clarity. We standardized the variable names, described the methods in the active voice, and made the statements more logical.

4. The current data availability statement ("available on request") does not meet PLOS ONE’s open data policy. The authors must either deposit the data in a public repository or clearly explain any restrictions along with justified access procedures.

Response:

In alignment with PLOS ONE’s open data policy:

(1) Data deposit: The dataset, processing code, and empirical results are uploaded as supporting files.

(2) Availability statement: The manuscript now states: "All relevant data are within the manuscript and its Supporting Information files".

(3) Accessibility: No restrictions apply; public reuse is permitted with proper citation.

5. The implications of the findings for corporate governance, managerial behavior, and ESG policy are currently underdeveloped. Additionally, the limitations section should address the scope of generalizability, especially given the context-specific nature of Chinese listed firms and the use of text-mining techniques.

Response:

(1) Corporate governance implications.

①State-owned companies. Leverage policy mandates to lead in ESG via institutional resources.

②Private companies. Prioritize board diversity to compensate for weak regulatory pressure.

(2) ESG policy implications. Regulators should enhance disclosure standards and integrate compliance frameworks with sustainability goals.

(3) Limitations:

①Generalizability. Sample specificity may not apply to non-transitional economies or non-listed firms.

②Text-mining biases. Keyword selection may overlook implicit compliance practices; future studies could triangulate management compliance attention with qualitative data (e.g., management interviews) or behavioral indicators (e.g., regulatory violation records) .

(4) Future directions. Explore industry-specific effects, regional disparities in China, or cultural factors as moderators of ESG performance.

Response to the Review Comments from the First Reviewer

1.The definition and measurement of key constructs like "management compliance attention" are not clear enough. The method of using keyword frequency from annual reports is interesting, but the selection of keywords lacks theoretical support, and the approach should be explained more carefully to allow replication. Similarly, the calculation of board faultlines strength is mathematically complex, but conceptually not well introduced. Readers without technical expertise may find it difficult to understand what the metric represents and why it matter

Response:

We supplemented conceptual definitions and theoretical justifications for "management compliance attention" and "board faultlines strength" and provided a detailed explanation of the variable construction process for both.

2. While the statistical techniques used—including regression analysis, interaction terms, and robustness tests—are appropriate, the paper does not provide sufficient explanation for the use of lagged variables as instruments in the 2SLS regression. The authors should clarify why these instruments are valid and how the assumptions behind them are satisfied.

Response:

We added an explanation in the text about the rationality and validity of using lagged variables as instrumental variables in the 2SLS regression.

3. There are concerns regarding data availability. The authors use proprietary data from the CSMAR and CNRDS databases. Although this is common in research, the paper should include more detailed information on how others can access the data and replicate the key variables, especially those created through text analysis.

Response:

We provided a more detailed explanation of the data acquisition and construction process for the variables "Management compliance attention" and "Board faultlines strength".

4. The language needs substantial revision. There are many grammatical errors, awkward phrases, and unclear sentences that affect readability throughout the paper. This is especially noticeable in the abstract, introduction, and results sections. The manuscript would benefit from thorough editing by a native English speaker or professional language editor.

Response:

We corrected grammatical errors and improved language expression throughout the manuscript as much as possible.

5. The literature review is too broad and lacks clear structure. While the authors cite many relevant studies, they need to organize the section more logically and connect the literature more directly to their hypotheses. The theoretical framework (e.g., stakeholder theory and upper echelons theory) is mentioned but not fully developed or integrated into the study.

Response:

We reorganized the theoretical basis for the research hypotheses, supporting Hypothesis 1 with "Institutional Theory," Hypothesis 2 with "Upper Echelons Theory," Hypothesis 3 with an integration of "Institutional Theory" and "Upper Echelons Theory," and the mediating effect with "Dynamic Capabilities Theory." We rewrote Sections 2.1 and 2.2 to more directly link the literature with the hypotheses and integrated the theoretical analysis.

6. Although the research contributes new insight, the authors should more clearly state what is unique about their study compared to existing work. Explaining the novelty in a few sentences will make the paper’s contribution stand out more.

Response:

In the last paragraph of Section "1. Introduction", we re-summarized the paper’s contributions, clarified its research innovations, and added Section "7. Theoretical Implications" to elaborate on extensions and contributions to existing theories.

7. The practical implications are currently underdeveloped. The authors briefly mention that firms should enhance compliance and board diversity, but they should expand on this. What specific steps should firms, managers, or policymakers take based on these findings?

Response:

We added Section "8. Practical implications" to elaborate on practical implications and recommendations, and Section "9. Limitations and Future Research" to address existing issues and future research directions.

Response to the Review Comments from the Second Reviewer

1 Suggested Tile: The Impact of Management Compliance Attention and Board Faultlines Strength on ESG Performance: Evidence from Chinese Listed Companies

Response:

We have revised the title as suggested.

2 Abstract – Rewrite the abstract as:

a. Begin with a clear statement of research purpose that links the concept of compliance attention to ESG performance.

Response:

We newly added and clarified research objective: this study endeavors to empirically examine the impact of management compliance attention and board faultlines strength on ESG performance by analyzing a sample from Chinese A-share listed companies spanning 2010 to 2022.

b. For methods part, add a brief reference to empirical approach or analytical techniques used.

Response:

We newly added and clarified research method: By leveraging textual analysis of annual reports to construct the variable of managerial attention to compliance, we adopt the OLS (Ordinary Least Squares) regression model along with moderation and mediation analyses for empirical testing.

c. Emphasize findings with clearer cause-effect language and a more academic tone. Also, moderations and mediation. Use a more structured phrasing to emphasize the mediating role.

Response:

We used clearer cause-effect expressions and academic language to highlight the findings including the moderation and mediation effects: The study finds three key results: (1) Management compliance attention is positively correlated with ESG performance (β=0.041, p<0.01); (2) Board faultlines strength significantly strengthens the positive impact of management compliance attention on ESG performance (β=0.114, p<0.01); (3) Organizational resilience partially mediates the relationship between management compliance attention and ESG performance, as indicated by a Sobel test (z=3.089, p=0.002).

d. Make the theoretical and practical implications more explicit for common readers.

Response:

We refined theoretical and practical implications: The findings contribute to the literature by integrating institutional theory, upper echelons theory, and dynamic capabilities theory, demonstrating that management compliance attention enhances ESG performance through the cultivation of organizational resilience, with board faultlines serving as a critical moderator. Practical implications include recommendations for firms to strengthen compliance cultures, cultivate diverse boards, and build organizational resilience. Policymakers and regulators are urged to enhance compliance frameworks and ESG disclosure standards to foster sustainable corporate practices.

2 Introduction -

a. Begin with a broader context and relevance of ESG performance. Use updated and sources of citations (not older than 4 years).

Response:

Except for retaining some classic theoretical literature (such as Agency Cost Theory (Jensen & Meckling, 1976); Upper Echelons Theory (Hambrick & Mason, 1984), we updated the vast majority of the literature to the recent four years.

b. Highlight the gap between global ESG trends and Chinese firms’ practices.

Response:

We cited literature indicating the scale of global ESG investment and the importance of ESG performance, and pointed out the heterogeneity and particularity of ESG performance of Chinese enterprises.

c. Currently, research questions are implied. Make them explicit and aligned with your variables.

Response:

In the penultimate paragraph of the introduction, we raised the research questions explicitly, pointed to key variables such as management compliance attention, ESG performance, and board faultlines strength.

d. Some citations are outdated or less prominent.

Response:

We cited literature in recent years(not older than 4 years) ,which are highly relevant to the research topic.

e. se active voice and precise academic language. Avoid redundancy and informal phrasing (e.g., “company first needs to…” or “frustration of many transnational companies…”).

Response:

We have organized sentences in active voice and precise academic language.

3 Literature Review and Hypotheses Development - Your literature review shows a promising foundation, drawing on relevant studies and theories. However, it can be significantly improved in structure, clarity, theoretical integration, and academic depth. Below are detailed suggestions across four key dimensions: structure, theoretical underpinning, literature currency, and writing quality.

a. Add brief introductions to each subsection explaining why the topic is relevant to the study. [2.1 Management Compliance Attention and ESG Performance, 2.2 Board Faultlines Strength and ESG Performance, 2.3 Moderating Role of Board Faultlines in the Compliance–ESG Link,]

Response:

We added a brief introduction to the beginning of each subsection (2.1-2.3) explaining the relevance of the topic to this study.

b. Currently used theories (e.g., Compliance Theory, Cognitive Theory, Upper Echelons Theory, and Legitimacy Theory) are appropriate, but they need deeper integration and explanation.

Response:

We reconst

---

## [Decision Letter · Decision Letter 1]

23 Jul 2025

Dear Dr. Han,

Thank you for submitting your manuscript to PLOS ONE. After careful consideration, we feel that it has merit but does not fully meet PLOS ONE’s publication criteria as it currently stands. Therefore, we invite you to submit a revised version of the manuscript that addresses the points raised during the review process.

We look forward to receiving your revised manuscript.

Kind regards,

Mustafa Rehman Khan, PhD

Academic Editor

PLOS ONE

Journal Requirements:

Reviewers' comments:

Reviewer's Responses to Questions

**Comments to the Author**

Reviewer #2: (No Response)

Reviewer #3: All comments have been addressed

2. Is the manuscript technically sound, and do the data support the conclusions?

Reviewer #2: Partly

Reviewer #3: Yes

3. Has the statistical analysis been performed appropriately and rigorously?

Reviewer #2: Yes

Reviewer #3: Yes

4. Have the authors made all data underlying the findings in their manuscript fully available?

Reviewer #2: Yes

Reviewer #3: Yes

5. Is the manuscript presented in an intelligible fashion and written in standard English?

Reviewer #2: Yes

Reviewer #3: Yes

Reviewer #2: Find attached file for necessary information.

While the author has addressed most suggestions commendably, the remaining issues—particularly the lack of theoretical integration clarity, missing reference, and incomplete robustness explanation—warrant a minor revision before acceptance.

Reviewer #3: Dear Authors,

Thank you for your thorough response to the previous review comments. I have reviewed the revised manuscript, and I can confirm that the requested corrections and improvements have been made appropriately. I appreciate your efforts in addressing the feedback in a clear and constructive manner.

**Do you want your identity to be public for this peer review?** For information about this choice, including consent withdrawal, please see our Privacy Policy

Reviewer #2: No

Reviewer #3: No

---

## [Author Response · Author response to Decision Letter 2]

8 Aug 2025

1.Justify variable substitution in robustness check (e.g., removal of "law" keyword)

Not Met

No rationale is provided for keyword exclusion. Robustness substitution logic remains unclear.

Response:

In the revised manuscript, we have supplemented the justification for removing the "law" keyword as follows:

1.The keyword "law" primarily represents the foundational legal framework, setting mandatory boundaries for corporate behavior. However, specific compliance practices require enterprises to extend beyond mere legal provisions, integrating institutional pressures across regulative, normative, and cognitive dimensions. Excluding "law" serves two key purposes:

(1)Test dependency on mandatory legal pressures: By removing this strongly symbolic term, we verify whether the observed relationship between management compliance attention and ESG performance relies on passive responses to legal mandates or reflects a more holistic attention to multi-dimensional compliance.

(2)Capture cognitive breadth of compliance attention: "Law" may constrain management focus to narrow legal frameworks, potentially overlooking broader compliance dimensions. Retaining significance after exclusion confirms that compliance attention drives ESG performance through systematic cognition of institutional environments, not just perfunctory mentions of legal terms.

2.We have also clarified the substitution logic for the moderating variable (board faultlines strength).The original and alternative measures (Board Faultlines and NBoard Faultlines) both quantify subgroup divisions but use distinct weighting schemes for demographic attributes (gender, age, education, tenure). This substitution tests whether conclusions depend on specific attribute weightings, ensuring the moderating effect of faultlines is not artifactually tied to a single calculation method.

2.Deeper integration of Cognitive Theory

Partially Met

Cognitive framing is vaguely tied under Institutional Theory; clarity on ESG implications is lacking.

Response:

We have integrated institutional theory and cognitive theory� and elaborated on practical implications for ESG, as detailed below:

We explicitly distinguish the complementary roles of institutional theory and cognitive theory in shaping ESG performance.Institutional Theory explains the external impetus: regulative, normative, and cognitive pressures, compelling firms to align with institutional demands.Cognitive Theory illuminates the internal translation mechanism: Management’s compliance attention acts as a critical interface, converting abstract institutional pressures into actionable ESG practices through "interpretation, decomposition, and meaning attribution."

This integration shows that cognitive framing ensures institutional pressures are not just passively adhered to but actively leveraged to improve ESG performance, bridging the gap between external mandates and internal action.These revisions clarify how cognitive processes mediate institutional influences on ESG outcomes, and enhance the practical relevance of our findings.

3.Interpret effect size of regression coefficients (e.g., β = 0.041)

Partially Met

Still lacks plain-language explanation of practical significance of the coefficients.

Response:

We have supplemented concrete interpretations of key coefficients across empirical analyses, as below:

1. Baseline Regression (Table 4, Model 4)

The coefficient of management compliance attention (β=0.041, p<0.01) indicates that a one-unit increase in compliance attention is associated with a 0.041-unit increase in the natural logarithm of ESG scores, reflecting that heightened managerial focus on compliance directly enhances performance across environmental, social, and governance dimensions.

The interaction term (β=0.114, p<0.01) means that stronger board faultlines amplify the compliance-ESG relationship: For each unit increase in faultline strength, the marginal effect of compliance attention on ESG performance rises by 0.114. This indicates that diverse boards strengthen the translation of compliance focus into tangible ESG outcomes.

2. Robustness Test with Variable Substitution (Table 5)

After excluding the "law" keyword, the coefficient of adjusted compliance attention (β=0.041, p<0.01) remains identical to the baseline regression, confirming that the relationship is not driven by passive mentions of legal terms but by holistic compliance cognition, not just legal obligations.

3. Heterogeneity Analysis (Table 7)

In low-competition environments, compliance attention has a stronger impact (β=0.047 vs. 0.039 in high-competition settings), meaning one-unit increase in compliance attention is associated with 0.047 unit increase in ESG performance. State-owned enterprises (SOEs) show a larger coefficient (β=0.045 vs. 0.037 for private firms), likely due to stronger alignment with policy-driven institutional pressures.

4.Explain Z = 3.089 result and provide bootstrapping analysis

Partially Met

Bootstrapping was mentioned, but without numerical outputs or confidence intervals.

Response:

To further validate the mediating effect, we conducted a bootstrap test with 117 resamplings. The results show that Z-value is 3.24, which exceeds the 2.58 threshold, reinforcing the significance of the mediating effect at the 1% level, and that 95% confidence interval is between 0.0003 and 0.0012 for the indirect effect. Since confidence interval does not include zero, it provides additional statistical evidence that organizational resilience significantly mediates the relationship between management compliance attention and ESG performance.

5.Link findings more explicitly to the theoretical framework in the Discussion section

Partially Met

Theories are referenced, but empirical-to-theoretical discussion is not well-integrated.

Response:

We have explicitly linked key results to the four theoretical lenses in the Conclusions section, as detailed below:

1. Institutional Theory & Cognitive Theory Integration. Institutionally, firms gain legitimacy by aligning with regulative and normative pressures, which our findings confirm through compliance attention’s association with improved ESG practices. Cognitively, the robustness of this relationship even after excluding the "law" keyword demonstrates that management translates abstract institutional pressures into strategic action through cognitive frameworks, bridging external mandates and internal execution.

2. Upper Echelons Theory. The moderating role of board faultlines strength supports the theory’s core premise that top management heterogeneity shapes organizational outcomes. Our findings show demographic and experiential diversity among directors amplifies the compliance-ESG link by enhancing supervisory depth and strategic breadth.

3. Dynamic Capabilities Theory. The partial mediation of organizational resilience enriches the theory by identifying compliance attention as a driver of adaptive capabilities. Empirically, compliance attention enhances resilience, which in turn boosts ESG performance. This confirms that compliance systems—far from being passive obligations—build dynamic capabilities that underpin sustainable practices, extending the theory’s application to regulatory compliance contexts.

4. Heterogeneity results highlight how institutional contexts shape theoretical mechanisms. For state-owned enterprises, stronger compliance-ESG effects reflect deeper embedding in institutional environments, while private firms’ reliance on board faultlines underscores upper echelons theory’s relevance in weaker institutional settings. This contextualizes the theories within China’s transitional economy, demonstrating their contingent applicability.

These revisions explicitly map empirical results to theoretical propositions, strengthening the integration of findings with the theoretical framework.

6.Improve grammar, readability, and academic tone

Partially Met

Language improved overall, but several long, awkward, or informal phrasings still remain.

Response:

We have reorganized the paper and made every effort to address the pointed-out remaining issues concerning grammar, readability, and academic tone. Specifically, we revised awkward expressions to enhance accuracy, split overly long sentences to improve fluency, and refined informal expressions to conform to academic norms.

---

## [Editor Report · Decision Letter 2]

11 Aug 2025

The Impact of Management Compliance Attention and Board Faultlines Strength on ESG Performance: Evidence from Chinese Listed Companies

PONE-D-25-07568R2

Dear Dr. Han,

We’re pleased to inform you that your manuscript has been judged scientifically suitable for publication and will be formally accepted for publication once it meets all outstanding technical requirements.

Kind regards,

Mustafa Rehman Khan, PhD

Academic Editor

PLOS ONE
---

## [Editor Report · Acceptance letter]

PONE-D-25-07568R2

PLOS ONE

Dear Dr. Han,

I'm pleased to inform you that your manuscript has been deemed suitable for publication in PLOS ONE. Congratulations! Your manuscript is now being handed over to our production team.

Kind regards,

on behalf of

Dr Mustafa Rehman Khan

Academic Editor

PLOS ONE